# Hematological and physiological responses in polo ponies with different field-play positions during low-goal polo matches

Kanokpan Sanigavatee[1,2☯], Chanoknun Poochipakorn[1,2☯], Onjira Huangsaksri[1,2☯], Thita Wonghanchao[1,2☯], Mona Yalong[3‡], Kanoknoot Poungpuk[3‡], Kemika Thanaudom[3‡], Metha Chanda[2,4]*

1 Veterinary Clinical Study Program, Faculty of Veterinary Medicine, Kasetsart University, Kampaeng Saen Campus, Nakorn Pathom, Thailand, 2 Department of Large Animal and Wildlife Clinical Science, Faculty of Veterinary Medicine, Kasetsart University, Kampaeng Saen Campus, Nakorn Pathom, Thailand, 3 Veterinary Science Program, Faculty of Veterinary Medicine, Kasetsart University, Kampaeng Saen Campus, Nakorn Pathom, Thailand, 4 Thailand Equestrian Federation, Sports Authority of Thailand, Bangkok, Thailand

☯ These authors contributed equally to this work.
‡ MY, KP and KT also contributed equally to this work.
* fvetmtcd@ku.ac.th

**Data Availability Statement:** The HRV data supporting this study's findings are available at https://doi.org/10.6084/m9.figshare.24785247.v1.

## Abstract

Strenuous exercise in traditional polo matches creates enormous stress on horses. Hematological and physiological measures may vary across different field-play positions. This study aimed to investigate the effort intensity and the impact of exertion on hematology and heart rate variability (HRV) in polo ponies with different positions. Thirty-two ponies, divided equally into eight teams, were studied. Each comprises forwards (number 1), midfielders (numbers 2 and 3), and defenders (number 4). Team pairs played the first chukka in four low-goal polo matches. Percent maximum heart rate (%HR_max), indicating ponies' effort intensity, was classified into five zones, including zones 1 (<70%), 2 (70–80%), 3 (80–90%), 4 (90–95%) and 5 (>95%). Hematological and HRV parameters were determined before, immediately after, and at 30-minute intervals for 180 minutes after chukkas; HRV variables were also obtained during warm-up and exercise periods. Results indicated that the number two ponies spent more time in zone 4 ($p < 0.05$) but less in zone 2 ($p < 0.01$) than the number four ponies. Cortisol levels increased immediately and 30 minutes afterward ($p < 0.0001$ for both) and then returned to baseline 60–90 minutes after exertion. Other measures (Hct, Hb, RBC, WBC, neutrophils, and CK enzyme) increased immediately ($p < 0.0001$ for all) and lasted at least 180 minutes after exertion ($p < 0.05$–0.0001). HRV decreased during the chukka until approximately 90 minutes afterward ($p < 0.05$–0.0001). The stress index increased during the chukka and declined to baseline at 60 minutes in number 1–3 ponies but lasted 90 minutes in those at number four. Effort intensity distribution differed among field-play positions. Decreased HRV indicated reduced parasympathetic activity during exercise, extending to 90 minutes after exertion in polo ponies. Defenders seem to experience more stress than those in other positions.

**Funding:** The author(s) received no specific funding for this work.

**Competing interests:** The authors have declared that no competing interests exist.

## Introduction

Horses can perform high-intensity exercise due to their capacity to increase oxygen consumption, extraction, and cardiac output during explosive efforts such as escaping predators [1]. However, horses utilized for sports may face potentially health-threatening conditions involving exogenous and endogenous stressors during their careers [1]. Stress following involuntary activities in equestrian sports could lead to reduced performance, homeostasis disturbance, and medical problems in the body systems involved in those efforts [2, 3].

Equestrian polo is the only equestrian sport where team pairs play against one another [4]. Each team comprises four combinations in which the player's abilities are classified from –2 (low performance) to +10 (high performance). The total handicap of a team's players is summed to account for the team's handicap (goals) [4, 5]. Played on about a 10-acre (4.05 ha) grass field, a game consists of four, five, and six chukkas in low, medium, and high-goal competitions [6].

Polo pony is the term used to describe horses in polo games, referring to their agile type rather than their size [5]. In general, their training comprises three patterns, including 1) vereo, aerobic training in which up to five polo ponies are simultaneously exercised by walking, trotting, and light cantering for approximately 30 minutes; 2) taqueo, field training for a specific activity in the polo game; and 3) training match play for polo games [4]. Concerning these training patterns, exercise in equestrian polo aligns with a combined aerobic/anaerobic energy pathway [5]. Polo horses experience rapid speed acceleration, pauses, and swift direction changes throughout the seven minutes of each chukka [4, 5, 7]. Accordingly, this could impair homeostasis and compromise horse welfare during the match, especially in unfit horses.

Changes in hematological parameters have been reported in response to the exercise regimen in equestrian polo. Red [4, 5] and white blood cell [5] parameters increase immediately after polo matches. Furthermore, creatine kinase (CK) enzyme and lactate levels are also modulated [4, 8, 9], and blood pH and total carbon dioxide decrease, coinciding with increased base excess and lactate levels; these changes reflect that exertion during polo matches is at high-intensity [4, 8]. Measured heart rate (in terms of percentage of maximum heart rate, % $HR_{max}$) is frequently used to determine the cardiovascular demand and exercise intensity in sport horses in different disciplines [10, 11], including polo ponies [7, 12]. One study showed that, during a low-goal polo match, ponies spent most of the time at less than 90% $HR_{max}$ but a few minutes above 90%, suggesting that exercise during the match created moderate-to-high stress on the cardiovascular system [7].

Changes in various parameters can be studied to determine horses' stress responses, including cortisol levels [13–15], behavior [16–18], and heart rate variability (HRV) [19–22]. It is well-accepted that levels of cortisol, a glucocorticoid hormone generated by the adrenal cortex, are frequently used to monitor stress levels in multiple conditions, such as road transportation [14, 23], stress-related disease [24], and during equestrian competition [13, 25, 26]. It has been reported that horses participating in eventing experience higher cortisol levels than those in jumping and dressage [15]. In contrast, endurance horses have the highest cortisol levels post-exercise, compared to the cross-country phase of 3-day events, trotting and galloping races, and jumping [27]. Hence, stress levels differ among equestrian sporting disciplines according to exercise intensity and the demands of competing in specific sports [2, 13, 27].

As an alternative to cortisol, HRV has also been used as a non-invasive marker of autonomic and physiological responses that can reflect stress levels and welfare [13, 14, 22, 28, 29]. HRV is a continuous measure of change in time differences between consecutive heartbeats during the cardiac cycle. Fluctuations in the interval between heartbeats are influenced by the

interplay between sympathetic and parasympathetic (vagal) components that act on the sino-atrial node of the heart in conjunction with respiration and neurohormonal factors released into the blood circulation [30–33]. The irregular time interval between sequential heartbeats benefits the adaptability and flexibility of the body to cope with difficult situations [29, 33]. HRV appears to be a reliable indicator of autonomic responses in horses in several conditions, including specific veterinary protocols [34, 35], transportation [14, 36, 37], exercise training [20, 22, 38], and exercise during equestrian events [13, 39, 40].

Several HRV measures indicate the sympathetic and parasympathetic (vagal) activities in response to external stimuli. Decreased verified beat-to-beat (RR) interval, the root mean square of successive RR interval differences (RMSSD), and the standard deviation of the Poin-caré plot perpendicular to the line of identity (SD1) reflect reduced vagal activity during short and medium-distance road transportation in horses [41]. On the contrary, increased RMSSD following aromatherapy indicated a shift toward vagal action [19]. Concerning equestrian competition, differences in RR interval, RMSSD, the relative number of consecutive RR interval pairs that differ by more than 50 msec (pNN50), SD1, and VLF values were observed in horses jumping different fence heights [40]. A reduction in RMSSD was also documented in horses participating in equestrian sports [13, 21, 42]. Hence, HRV is a helpful indicator for determining autonomic responses in horses.

Equestrian polo shares a similar exercise pattern with human sports such as soccer, including sprinting interspersed with body contact and swift direction changes [43]. Thus, knowledge of physical and physiological responses may be transferred from human team sports to monitoring bodily responses in polo ponies. In competitive soccer matches, it has been reported that wingbacks perform more maximum exercise intensity than forwards [44]. Moreover, from a young age, outfielders (forwards, midfielders, and defenders) display higher $VO_2$ max, mean aerobic power, and sprint performance than goalkeepers [45]. Hence, different player positions experience distinct exercise intensities and physiological demands during soccer games.

Although physiological responses have been reported in soccer players with different field-play positions, similar responses in ponies across different positions are scarce. Therefore, this study aimed to investigate the impact of traditional exercise regimens in equestrian polo on effort intensity distribution, cortisol levels, hematology, and autonomic responses in polo ponies with different field-play positions.

## Methods

### Horses

We studied 32 healthy Argentine polo ponies (nine geldings and 23 mares, aged 12.5 ± 2.4 years, weighing 413 ± 10 kg) from the Thai Polo and Equestrian Club, Pattaya, Thailand. They had been regularly trained according to the traditional polo training method [12] and played two to four chukkas weekly during polo tournaments. They were housed separately in a 4 x 5 m stable within barns accommodating up to 40 horses. Daily feeding consisted of 2 kg of commercial pellets (provided across two meals) and 15 kg of hay hung permanently. Ponies had free access to tap water in their stables. They also spent approximately two hours daily in a paddock, except on match days. The inclusion criteria for recruiting the animals in this study were: 1) they had been regularly training and participating in polo games during the competitive season, and 2) no medical or surgical treatments were given before the experiment commenced. The polo ponies would have been excluded from the study if they suffered injuries or metabolic problems following a game. However, no pony was eliminated, and the data from all

32 polo ponies were utilized for quantitative analyses. The study protocol was approved by the Kasetsart University's Institute of Animal Care and Use Committee (ACKU65-VET-003).

## Experimental protocol

**Exercise regimen in a polo game.** The study was conducted in January 2022 during the Thai Polo and Equestrian Club tournament in Pattaya, Thailand. Thirty-two polo ponies were divided into eight teams, in which four player positions were deployed in a team, including number 1 (forward), numbers 2 and 3 (midfielders), and number 4 (defender). A team pair played against each other in four low-goal field plays outlined by the basic rules of polo [4]. The player's handicaps within the teams were 8.88 ± 2.42 goals. Since there were four chukkas, with a period of six and a half minutes per chukka, the ponies were assigned to participate only in the first chukka of each match for four consecutive days. The matches occurred in the late afternoon (15.00–17.00 h local time), during fine weather with an average humidity of 37.75 ± 2.75% and mean temperature of 35.60 ± 0.57˚C. The relevant sampling was conducted before, during, and after the first chukka.

**Hormonal and hematological analyses.** Hormonal and hematological variables were determined before the chukka, immediately afterward, and at 30-minute intervals for a further 180 minutes. In brief, 9 ml of whole blood was withdrawn from the jugular vein and divided into two parts. Three ml of blood was placed in potassium EDTA tubes for evaluating hematocrit (Hct), red blood cells (RBC), hemoglobin concentration (Hgb), white blood cells (WBC), neutrophils, and lymphocytes using an automated hematology analyzer (Advia®2120i; Siemens Healthineers, Erlangen, Germany). Another portion of the blood sample was placed in 6 ml clotting activator tubes to obtain serum for biochemical examination, including 1) cortisol level determination using a competitive chemiluminescent enzyme immunoassay (IMMU-LITE Analyzers, Siemens Healthineers, Erlangen, Germany) and expressed as nmol/L, and 2) evaluation of creatine kinase (CK) level using a Liquid NAC activated UV test (HUMAN Gesellschaft für Biochemica und Diagnostica mbH, Wiesbaden, Germany) and expressed as ukat/L.

**Heart rate and heart rate variability analyses.** *Data collection.* Ponies were equipped with portable Polar heart rate monitoring (HRM) device sets (Polar Electro Oy, Kempele, Finland), which are proven to provide reliable measures of HR and HRV analyses in horses [46–48]. The device set comprises a Polar heart rate equine belt for trotters, a heart rate sensor (Polar H10), and a sports watch for recording RR intervals (Polar vantage 2). In brief, the heart rate sensor was attached to the equine belt for trotters at the sensor pocket on the belt and then soaked in water to increase signal transmission. The soaked belt was fastened around the pony's chest, where the sensor was placed on the left side of the chest. The sensor was connected to the sports watch to record RR interval data 15 minutes before the chukka. The players wore sports watches for recording RR intervals during the competition, which were then left close to the ponies with whom the sensors were paired for 180 minutes after the competition.

*Data acquisition.* RR interval data from the sports watch was transferred to the Polar Flow program (https://flow.polar.com/) for HRV analysis. HRV variables were later computed from RR interval data using Kubios Premium software (Kubios HRV Scientific; https://www.kubios.com/hrv-premium/) and exported as the MATLAB MAT-file (S1 Fig). The premium software provides the automatic correction algorithm, which has been validated to correct artifacts and ectopic beats in the interbeat interval (IBI) data; in turn, this produces more accurate HRV variables than the standard version [49]. The Kubios HRV Scientific software also supports automatic noise detection to identify noise segments that distort various consecutive beat

detections in the IBI data. The automatic noise detection was set at a medium level in this study. Before analyzing HRV variables, smoothness priors were used to remove IBI time series nonstationarities. According to the user guideline manual, the cutoff frequency was fixed at 0.035 Hz (https://www.kubios.com/downloads/Kubios_HRV).

The HRV variables were as follows:

**Time domain results**: RR interval, HR, SDNN, RMSSD, pNN50, and stress index.

**Frequency domain results**: VLF band (by default 0–0.04 Hz), LF band (by default 0.04–0.15 Hz), HF band (by default 0.15–0.4 Hz), LF/HF ratio, total power spectrum, VLF/total power ratio, LF/total power ratio, HF/total power ratio and respiratory rate (RESP).

**Nonlinear results**: SD1 and SD2.

The HRV variables were determined 15 minutes before the match began, during the six and a half minutes of the first chukka, and at 30-minute intervals for 180 minutes after it.

**Time spent within effort intensity zones.** Real-time HR modulations during each chukka were derived from the Polar HRM device. The ponies' effort intensity during the chukka was estimated using the percent maximum HR (%HR$_{max}$), similar to previous reports in a competitive soccer match [44]. The %HR$_{max}$ was calculated using horses' reference maximum heart rate at 220 beats/min [(recorded HR x100)/220], according to previous work [12]. The computed %HR$_{max}$ of the ponies was then categorized into five zones, including zone 1 (<70% HR$_{max}$), zone 2 (70–80% HR$_{max}$), zone 3 (80–90% HR$_{max}$), zone 4 (90–95% HR$_{max}$), and zone 5 (>95% HR$_{max}$). The distributions of time spent within each zone were determined in each pony.

**Speed and distance covered.** The ponies' riding speed and distance covered were recorded by the Polar sports watches worn on the players' wrists throughout the chukka and exported as km/h and kilometers, respectively.

## Statistical analysis

HR and HRV data were analyzed using GraphPad Prism version 10.1.0 (GraphPad Software Inc, San Diego, USA). In analyzing two independent variables, the Greenhouse–Geisser correction was automatically selected to estimate an epsilon (sphericity) and correct for lack of sphericity before analyses. Two-way ANOVA with repeated measures was applied to evaluate the independent effects of field-play position and time, as well as the interaction effect of field-play position-by-time, on changes in cortisol level, hematological, HR, and HRV variables in response to the exertion during competition. Dunnett's *post-hoc* test was later implemented to assess between-group differences at given time points and within-group differences compared to the control (baseline values taken before competition). Tukey's *post-hoc* test was also applied when necessary. Two-way ANOVA was further used to determine the independent effects of field-play position and effort intensity zone (%HR$_{max}$) and their interaction on the time-spent distribution within zones. Tukey's *post-hoc* test was implemented to compare time spent in different effort zones within and between field-play positions.

Regarding the analysis of one independent variable, the data's normal distribution was verified by the Shapiro-Wilk test. Due to the data being normally distributed, an ordinary one-way ANOVA followed by Tukey's *post-hoc* test was employed to estimate differences in riding speed and covered distance among the ponies playing in different positions. The Kruskal-Wallis test, accompanied by Dunn's multiple comparisons tests, was alternatively applied to assess players' handicaps among different field-play positions because the data showed a non-normal distribution. Data were expressed as means ± SD; $p < 0.05$ was considered statistically significant.

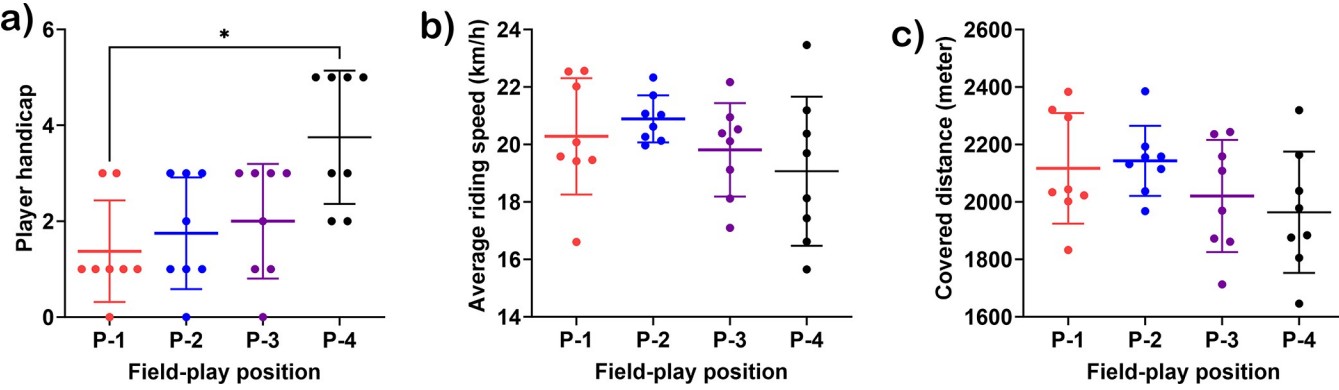

**Fig 1.** Player handicap (a), average riding speed (b), and covered distance of polo ponies during field-play competition. *indicates statistical significance between pairs of comparison at $p < 0.05$. **P-1**; horse played at number 1 position, **P-2**; horse played at number 2 position, **P-3**; horse played at number 3 position, and **P-4**; horse played at number 4 position.

## Results

### Player's handicap, riding speed, and distance covered

The average handicap of the number 4 players was higher than that of the number 1 players ($3.75 \pm 1.39$ vs. $1.38 \pm 1.06$, $p = 0.0170$) (Fig 1A). The average riding speed and covered distance did not differ among the field play positions (Fig 1B and 1C).

### Time spent within different effort zones

There was an interaction between field-play position and effort intensity zone (%$HR_{max}$) ($p = 0.0497$) and the separate effect of effort intensity zone ($p < 0.0001$) on the distribution of time spent within the zones. The ponies at the number 1 position spent more time in zone 3 than in zones 1 ($p < 0.05$), 4 ($p < 0.05$), and 5 ($p < 0.01$). The ponies at the number 2 position spent more time in zone 3 than in zones 2 and 5 ($p < 0.05$ for both) and in zone 4 more than zone 5 ($p < 0.01$). The time spent in different zones by the number 3 ponies did not vary. The number 4 ponies spent more time in zones 2 and 3 than both zone 4 ($p < 0.05$ for both) and zone 5 ($p < 0.001$ for both). The number 4 ponies spent more time in zone 2 ($p < 0.01$) and less time in zone 4 ($p < 0.01$) compared with the number 2 ponies (Fig 2).

### Hormonal and hematological analyses

Since there was no interaction effect between group and time on changes in hormonal levels and hematological parameters, the parameters were subsequently evaluated as a pool, following the main effect of time. Cortisol levels increased immediately after the match and remained high until 30 minutes afterward ($p < 0.0001$ for both periods). After reducing to the baseline value at 60–90 minutes after the match, cortisol levels decreased to below the baseline at 120–180 minutes ($p < 0.0001$ for all periods) (Table 1).

CK levels increased immediately until 150 minutes after the match ($p < 0.05$–$0.0001$). At the same time, hematological parameters (including Hct, Hb, and RBC) increased immediately until 180 minutes afterward ($p < 0.0001$ for given periods of all variables). Even though WBC and neutrophil levels rose immediately until 180 minutes after competition ($p < 0.0001$, except for $p < 0.05$ at 60 minutes after competition for both variables), lymphocytes were unchanged throughout the study period (Table 1).

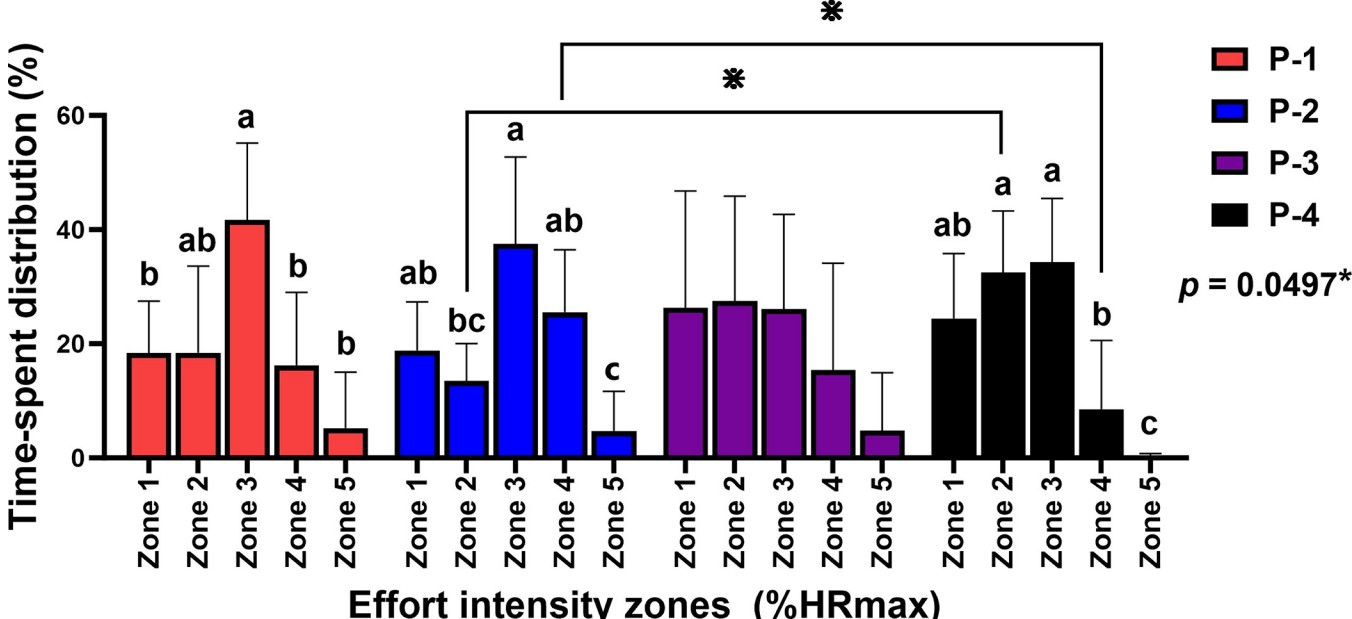

**Fig 2. Percentage time spent in each heart rate (HR) zone in polo ponies at different field-play positions 1–4 (P-1, P-2, P-3, and P-4).** *Indicates interaction *p*-value of analysis. Different letters (a, b, and c) indicate the statistical significance among HR zones within particular field-play positions. ✱ Indicates significant differences in HR zones between positions. Zone 1 (<70% HRmax: HR<154 bpm), Zone 2 (70–80% HRmax:155≤HR≤176 bpm), Zone 3 (80–90% HRmax: 177≤HR≤198 bpm), Zone 4 (90–95% HRmax: 199≤HR≤209 bpm), and Zone 5 (>95% HRmax: HR>210 bpm).

**Table 1. Biochemical and hematological parameters (mean ± SD) in polo ponies (n = 32) before and after low-goal polo competition.**

| Parameters | Before competition | After competition | | | | | | |
|---|---|---|---|---|---|---|---|---|
| | | Immediately | 30 min | 60 min | 90 min | 120 min | 150 min | 180 min |
| **Cortisol** (25–155 nmol/L) | 68.89± 10.19 | 115.53± 6.44[d] | 101.91± 4.7[d] | 75.70± 6.67 | 58.89± 6.12 | 45.14± 4.44[d] | 39.61± 0.85[d] | 31.67± 0.5[d] |
| **Creatine kinase (CK)** (0.17–5.83 ukat/L) | 4.06± 0.24 | 4.64± 0.36[d] | 4.70± 0.49[b] | 5.18± 0.63[b] | 5.87± 0.83[b] | 5.57± 0.67[b] | 5.48± 0.78[a] | 5.27± 0.83 |
| **Hematocrit (Hct)** (32–53%) | 32.29± 0.73 | 49.02± 1.04[d] | 37.61± 1.11[d] | 36.11± 1.77[d] | 37.82± 0.66[d] | 38.61± 1.12[d] | 37.79± 1.04[d] | 37.26± 0.5[d] |
| **Hemoglobin (Hb)** (110–190 g/L) | 117.28± 2.18 | 171.66± 3.28 [d] | 133.94± 3.4[d] | 128.56± 3.4[d] | 133.88± 1.7[d] | 136.34± 3.0[d] | 134.28± 2.8[d] | 132.63± 1.60[d] |
| **Red blood cell (RBC)** (6.80–12.90 x $10^{12}$/L) | 6.89± 0.11 | 10.25± 0.31[d] | 7.96± 0.25[d] | 7.63± 0.21[d] | 7.97± 0.21[d] | 8.14± 0.25[d] | 7.96± 0.30[d] | 7.86± 0.25[d] |
| **White blood cell (WBC)** (5.40–14.30 x $10^{9}$/L) | 6.48± 0.12 | 8.47± 0.31[d] | 7.23± 0.16[d] | 6.87± 0.35[a] | 7.61± 0.11[d] | 7.79± 0.27[d] | 7.80± 0.27[d] | 7.84± 0.28[d] |
| **Neutrophil (Neu)** (2.26–8.50 x $10^{9}$/L) | 4.30± 0.39 | 5.72± 0.39[d] | 4.80± 0.30[a] | 4.89± 0.54[a] | 5.50± 0.36[d] | 5.66± 0.05[d] | 5.69± 0.24[d] | 5.62± 0.24[d] |
| **Lymphocyte (Lymp)** (1.50–7.70 x $10^{9}$/L) | 2.02± 0.31 | 2.49± 0.25 | 2.23± 0.40 | 1.75± 0.25 | 1.88± 0.36 | 1.86± 0.21 | 1.79± 0.23 | 1.95± 0.17 |

**a**, **b**, and **d** indicate statistical differences of given time points compared to before competition values at $p < 0.05$, $p < 0.01$, and $p < 0.0001$, respectively.

### Heart rate and heart rate variability

**Time domain results.** Only time exerted effects on HR and most time domain HRV variables, except for stress index, which demonstrated a group-by-time interaction ($p = 0.0322$), in addition to the independent effect of time.

The minimum HR increased during the warm-up and competition period ($p < 0.0001$ for both periods). Despite a gradual reduction, the minimum HR remained higher than the baseline values until 180 minutes after the competition ($p < 0.0001$ for all periods) (Fig 3A). The maximum HR also rose dramatically during the warm-up and peaked during the competition ($p < 0.0001$ for both periods). It was still high 30 minutes after the competition ($p < 0.0001$) and then reduced considerably to the baseline 60–90 minutes afterward. Maximum HR decreased further, to below the baseline, at 120 ($p < 0.05$) and 180 minutes ($p < 0.05$) after the match (Fig 3B).

As expected, mean HR increased during the warm-up and competition ($p < 0.0001$ for both periods). After that, it reduced dramatically 30 minutes after the competition but remained higher than the baseline ($p < 0.0001$). Although it continued to decrease, it remained higher than the baseline 180 minutes after the match ($p < 0.05$) (Fig 3C).

In contrast to HR variables, the mean RR interval decreased during warm-up ($p < 0.00001$) and plunged to its lowest value during competition ($p < 0.0001$). There was a sharp rise in mean RR intervals 30 minutes ($p < 0.0001$) after the match. However, they then reached a

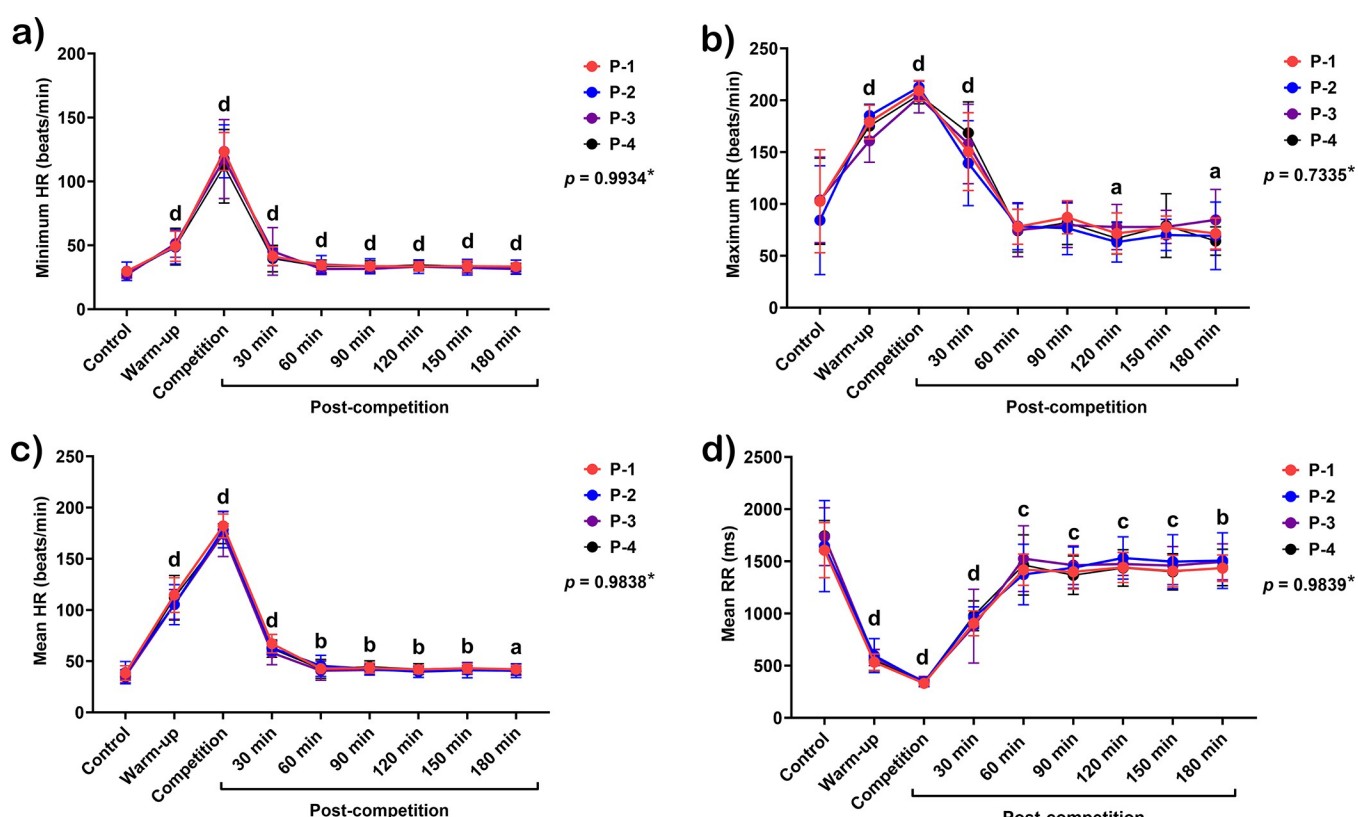

**Fig 3.** Modification in minimum HR (a), maximum HR (b), mean HR (c), and RR intervals (d) in response to exercise in polo matches. *Indicates the interaction *p*-value of all variables. **a, b, c,** and **d** letters indicate the statistical significance of each time point compared to the control at $p < 0.05$, $p < 0.01$, $p < 0.001$, and $p < 0.0001$, respectively. **P-1**; horse played at number 1 position, **P-2**; horse played at number 2 position, **P-3**; horse played at number 3 position, and **P-4**; horse played at number 4 position.

plateau and remained higher than the baseline 60–180 minutes after the competition ($p < 0.01$–$0.001$) (Fig 3D). SDNN, RMSSD, and pNN50 shared the same trend: they dropped dramatically during warm-up ($p < 0.0001$ for all variables) and then continued to decrease to their lowest values during the competition ($p < 0.0001$ for all variables). After competition, they increased markedly 30–90 minutes ($p < 0.001$–$0.0001$ for all variables) until reaching the baseline at 120 minutes. In addition, the pNN50 was higher than the baseline at 180 minutes after the match ($p < 0.05$) (Fig 4A–4C).

Since an interaction was observed between field-play position and time, the stress index differed among the ponies in the match. The stress index increased during warm-up (number 1 ponies, $p < 0.01$; number 2, $p < 0.05$; number 3, $p < 0.001$; number 4, $p < 0.01$) and peaked during competition periods ($p < 0.0001$ for number 1, 2, and 3 ponies, and $p < 0.001$ for number 4 ponies). It then reduced sharply 30 minutes after the competition (number 1 ponies, $p < 0.001$; number 2, $p < 0.05$; number 3, $p < 0.01$; number 4, $p < 0.001$). Although the stress index decreased to the baseline 60 minutes after competition in the ponies at numbers 1–3, it later reduced to the baseline 120 minutes after competition in those at number 4 ($p < 0.01$) (Fig 4D).

**Frequency domain results.** VLF, LF, and HF bands decreased during warm-up (VLF and LF, $p < 0.0001$; HF, $p < 0.01$) and the competition ($p < 0.0001$ for all variables) (Fig 5A–5C).

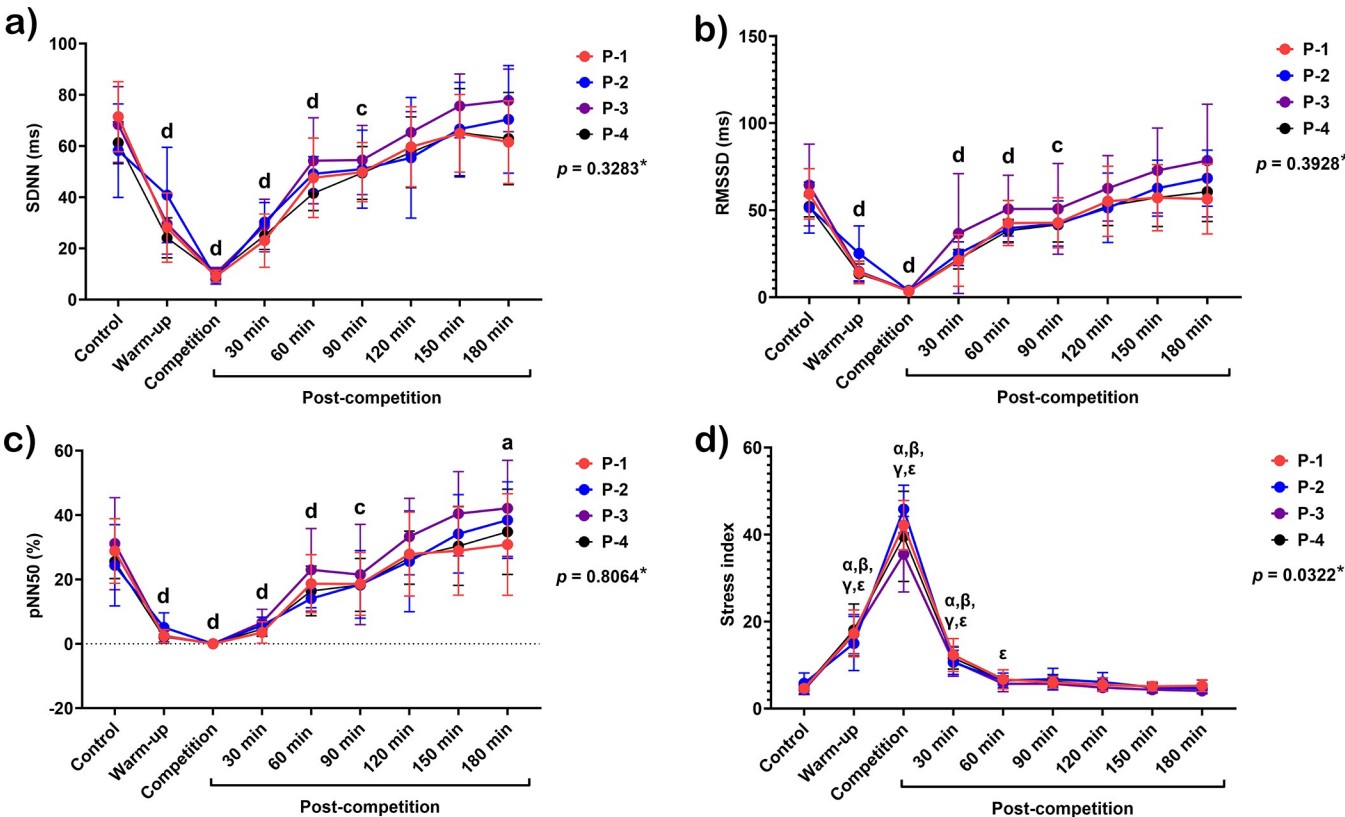

**Fig 4.** Time domain analysis of HRV in response to exercise in polo matches, including standard deviation of RR intervals (SDNN) (a), square root of the mean squared differences between successive RR intervals (RMSSD) (b), relative number of successive RR interval pairs that differ by more than 50 msec (pNN50) (c), and stress index (d). *indicates the interaction $p$-value of all variables. **a, b, c,** and **d** letters indicate the statistical significance of each time point compared to the control at $p < 0.05$, $p < 0.01$, $p < 0.001$, and $p < 0.0001$, respectively. **α, β, γ,** and **ε** indicate significant differences in each time point compared to their controls in P-1, P-2, P-3 and P-4, respectively. **P-1**; horse played at number 1 position, **P-2**; horse played at number 2 position, **P-3**; horse played at number 3 position, and **P-4**; horse played at number 4 position.

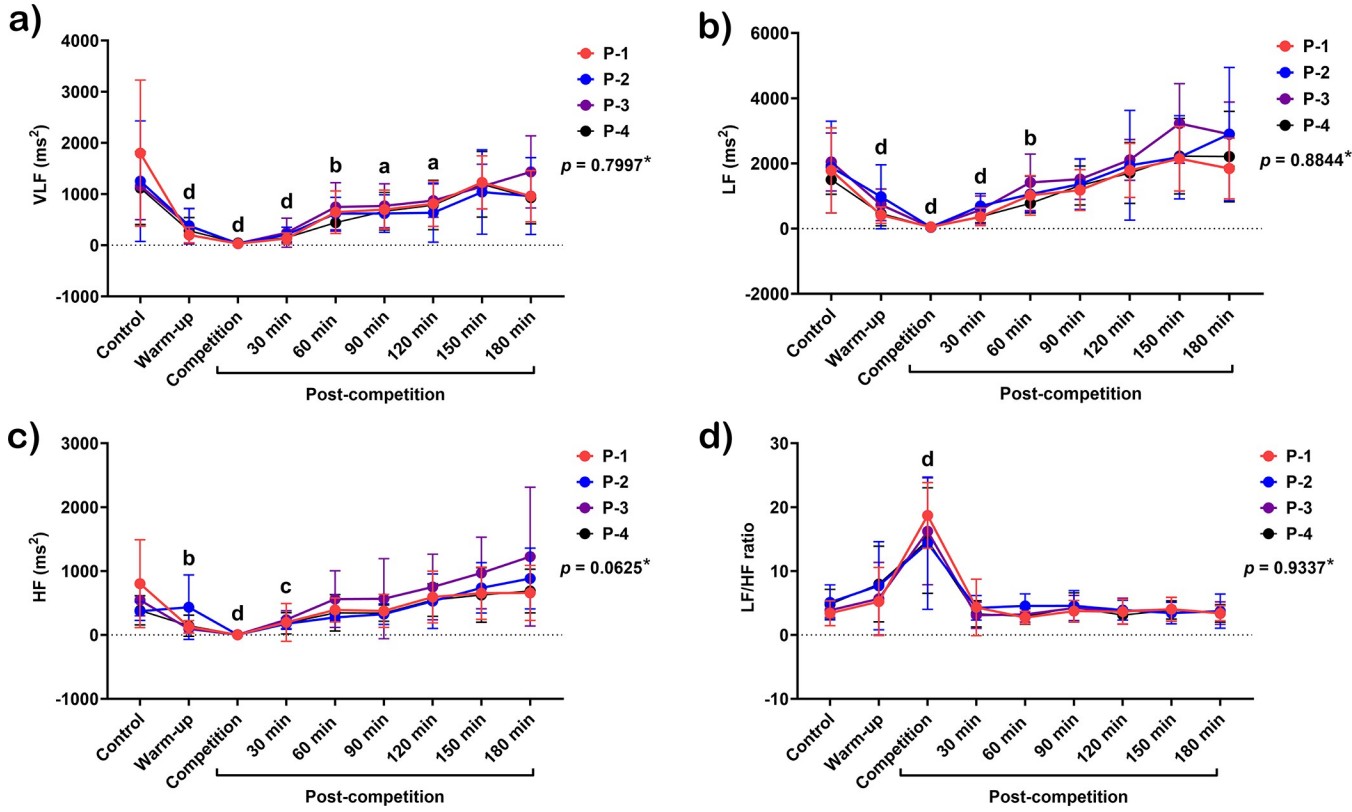

**Fig 5.** Changes in frequency domain analysis, including very-low-frequency band (VLF) (a), low-frequency band (LF) (b), high-frequency band (HF) (c), and LF/HF ratio (d) in response to exercise in polo matches. *indicates the interaction *p*-value of all variables. **a, b, c**, and **d** letters indicate the statistical significance of each time point compared to the control at *p* < 0.05, *p* < 0.01, *p* < 0.001, and *p* < 0.0001, respectively. **P-1**; horse played at number 1 position, **P-2**; horse played at number 2 position, **P-3**; horse played at number 3 position, and **P-4**; horse played at number 4 position.

VLF band rose 30–120 minutes (*p* < 0.05–0.0001) and returned to the baseline 150 minutes later (Fig 5A). LF band increased from 30 (*p* < 0.0001) to 60 minutes (*p* < 0.01) and reached the baseline 90 minutes later (Fig 5B). In the meantime, the HF band returned to the baseline earliest, at 60 minutes after the match (*p* < 0.001) (Fig 5C). LF/HF ratio increased during the match (*p* < 0.0001) and returned to the baseline 30 minutes later (Fig 5D).

The total power spectrum reduced from the warm-up period (*p* < 0.01), with the lowest value during the match (*p* < 0.0001), extending to 90 minutes afterward (*p* < 0.05–0.0001) (Fig 6A). The VLF/total power ratio increased during the match (*p* < 0.05), then reduced sharply 30 minutes (*p* < 0.0001) and 120–150 minutes afterward (*p* < 0.01–0.001) (Fig 6B). LF/total power ratio was unchanged throughout the study period (Fig 6C), while HF/total power ratio decreased during the match (*p* < 0.0001) (Fig 6D). RESP rose from the warm-up period to 30 minutes after the match (*p* < 0.0001 for all periods). However, it declined to the baseline during 60–90 minutes and then rebounded above the baseline 120–180 minutes afterward (*p* < 0.05–0.01) (Fig 7A).

**Nonlinear results.** SD1 and SD2 also decreased during warm-up and the match (*p* < 0.0001 for both periods of the two variables). They increased 30–90 minutes after the match (*p* < 0.001–0.0001 for both periods of the two variables) and reached the baseline 120 minutes afterward (Fig 7B and 7C). An increase in SD2/SD1 was observed during the warm-up and competition (*p* < 0.0001 for both periods) (Fig 7D).

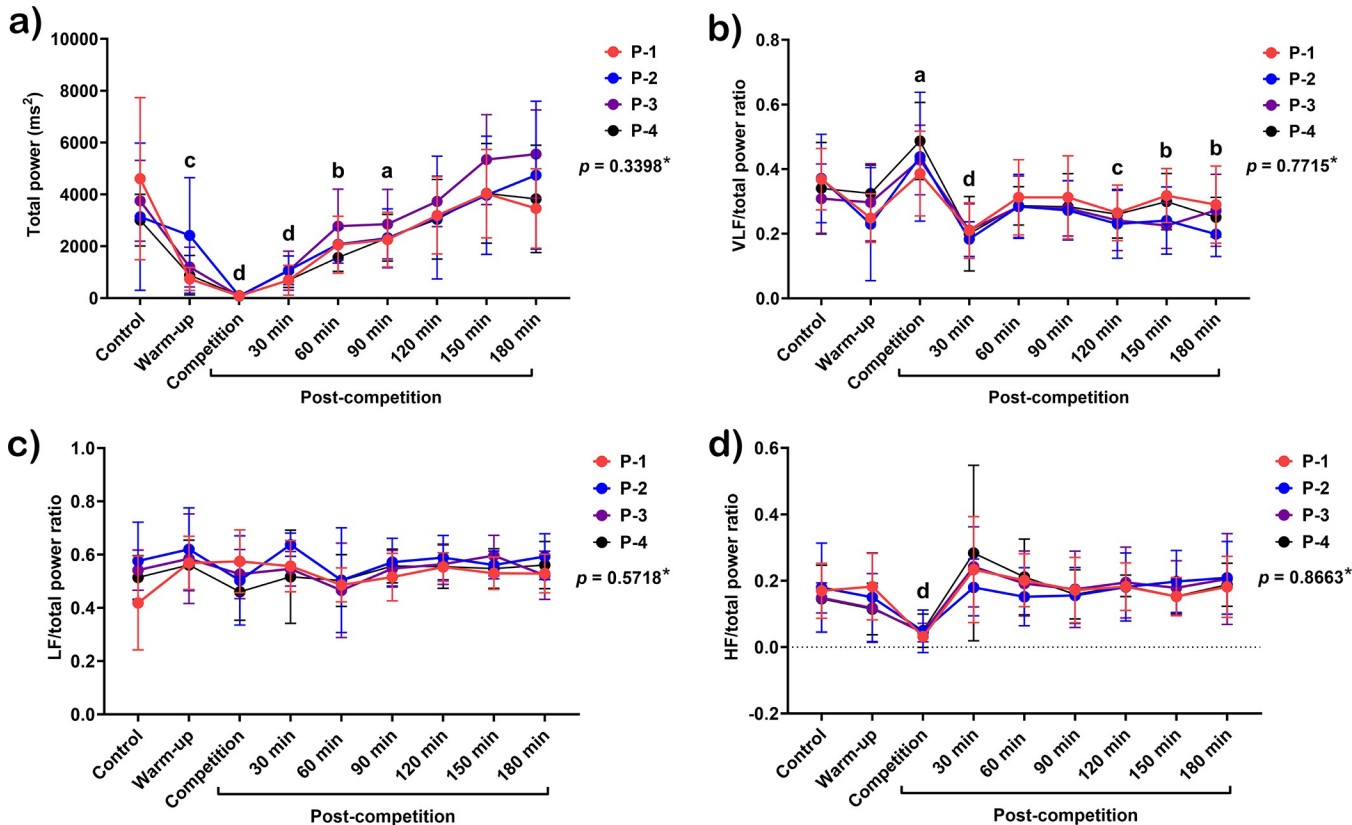

**Fig 6.** Total power spectrum (a), VLF/total power ratio (b), LF/total power ratio (c), and HF/total power ratio (d) in response to exercise in polo matches. *indicates the interaction $p$-value of all variables. **a, b, c**, and **d** letters indicate the statistical significance of each time point compared to the control at $p < 0.05$, $p < 0.01$, $p < 0.001$, and $p < 0.0001$, respectively. **P-1**; horse played at number 1 position, **P-2**; horse played at number 2 position, **P-3**; horse played at number 3 position, and **P-4**; horse played at number 4 position.

## Discussion

The present study determined the impact of a classic exercise regimen in a polo match on effort distribution, cortisol release, hematology, and heart rate variability. The significant findings from this study were: 1) number 2 ponies spent more time exercising in effort intensity zone 4 but less in zone 2 than number 4 ponies; 2) there were no differences in cortisol levels and hematological modulation among ponies with different field-play positions; 3) cortisol levels rose considerably during competition and returned to the baseline 60 minutes later; 4) Hct, RBC, Hb, WBC, neutrophils, and CK levels increased immediately and remained higher than the baseline up to 180 minutes after exertion; 5) mean HR increased considerably and remained higher than the baseline at 180 minutes; 6) HRV modulation did not differ among ponies in different field-play positions, except for the stress index, which declined later in the number four ponies than those in other field-play positions; and 7) HRV decreased to the lowest values during exercise, gradually increasing and returning to baseline 120 minutes afterward.

The results suggest that polo ponies at different field-play positions experienced different levels of effort intensity. A decreased HRV reflected a shift toward sympathetic dominance during warm-up, with a substantial sympathetic effect during the workout, which lasted approximately 90 minutes afterward. As indicated by the stress index modification, the number 4 ponies appeared to experience more stress than those in other positions. To our

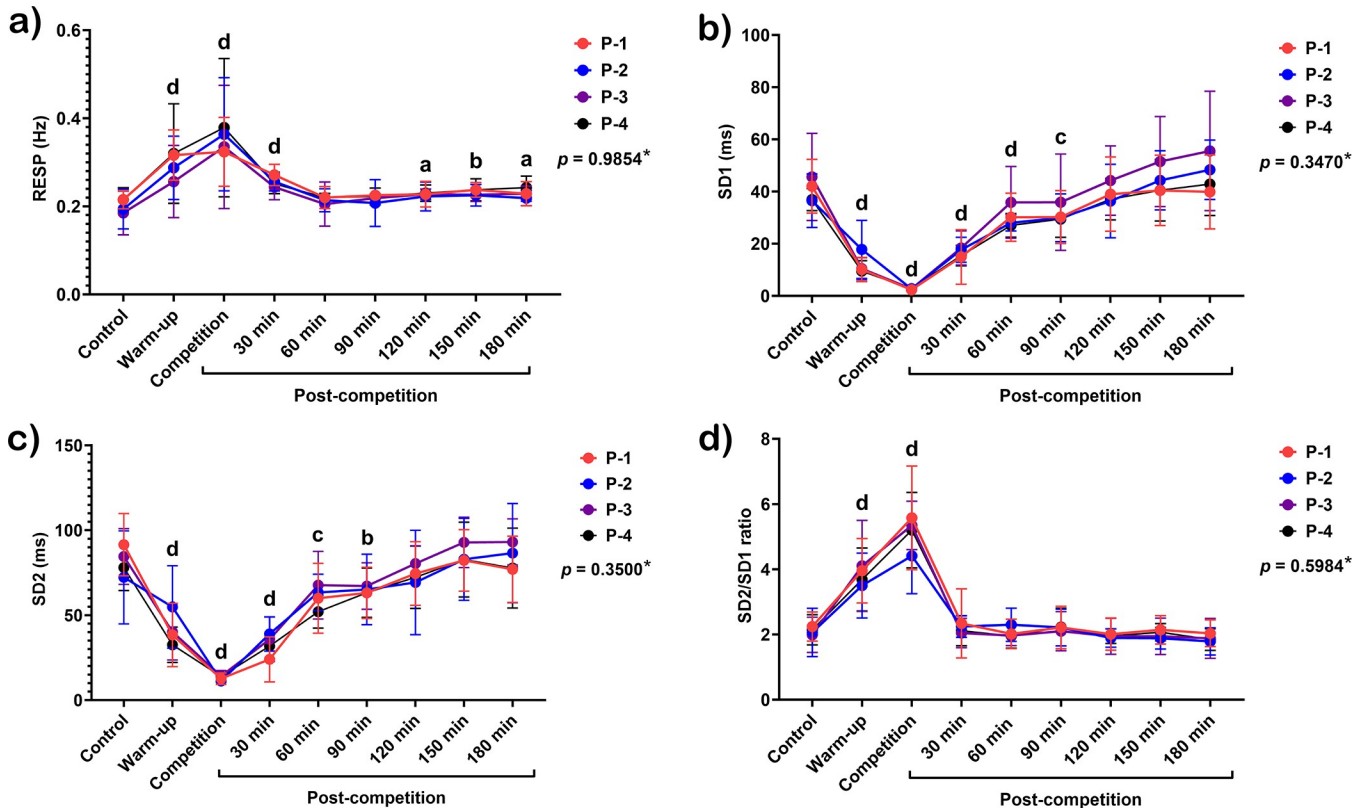

**Fig 7.** Changes in respiratory rate (RESP) (a), standard deviation of the Poincaré plot perpendicular to the line of identity (SD1) (b), standard deviation of the Poincaré plot along the line of identity (SD2) (c), and SD2/SD1 ratio (d) in response to exercise in polo matches. *Indicates the interaction $p$-value of all variables. **a, b, c**, and **d** letters indicate the statistical significance of each time point compared to the control at $p < 0.05$, $p < 0.01$, $p < 0.001$, and $p < 0.0001$, respectively. **P-1**; horse played at number 1 position, **P-2**; horse played at number 2 position, **P-3**; horse played at number 3 position, and **P-4**; horse played at number 4 position.

knowledge, this is the first study to report the physiological demands and autonomic responses in polo ponies with different field-play positions in low-goal polo matches.

In soccer, players' physical and physiological demands also vary across different positions [44, 45, 50]. Wingbacks spend more time in higher-intensity zones than forwards [44]. Wingbacks typically must participate in offensive action along with the forwards, alternating with a sprint back to cover opponents in a defensive system [44, 51]. In contrast, forwards and defenders spent more time with lower exercise intensity than other player positions because their specific task was scoring [44]. It is worth noting that specific training regimens, with appropriately adjusted workloads, can increase players' performance in different positions [44, 50]. Since equestrian polo and soccer have similar exercise patterns, differences in the positional demands are also expected in this equestrian sport.

In polo, each field-play position serves a distinct role. The number 1 position is usually filled by a player with relatively little experience, who is mainly responsible for scoring and neutralizing the opposite number 4. In contrast, the number 2 player acts as a scrambler who needs a keen eye and fast ponies to scrap for the ball. They may feed the ball to the number 1 player or score themselves. The number 3 player is the primary tactical player who provides the ball to the number 1 and 2 players, while the number 4 plays a crucial role in the defensive system to prevent the opposition from scoring [52]. In this study, the number 2 ponies spent more time in the higher effort zone and less time in the lower effort zones than those at

number 4. This seems consistent with the multifunctional role of the number 2 player [6]. In contrast, spending more time in lower effort zones in number 4 ponies may be consistent with their primary role in defense, undergoing multiple changes in direction and speed to prevent scoring [52]; consequently, the running speed of number 4 ponies may not reach the high values observed in those at number 2. Our data thus suggest that ponies' physical effort within the chukka varies among different field-play positions and that specific training may be required for each position to reach the required performance at the respective intensity levels.

Serum cortisol levels and hematological parameters changed during the matches, regardless of field-play positions. Cortisol levels are known to be modulated in horses participating in various equestrian sports [13, 26, 27], including polo [5, 53]. The modulation we observed was consistent with previous studies in horses [54, 55] and polo ponies specifically [5], showing a sharp increase in serum cortisol immediately, lasting 30 minutes, and returning to baseline 60 minutes after exertion. These results reflect the enormous stress on the ponies during high-intensity exercise in the chukkas. However, a further decrease in cortisol levels (below the baseline) at 120–180 minutes may result from the hormone's circadian rhythm during the late evening [22, 56–59].

The observed increase in CK enzyme levels parallels those of a previous report [5], demonstrating a rise after the chukkas but within the normal reference range of the enzyme. Increased CK levels are believed to accompany high-intensity exercise due to muscle micro-trauma, leading to increased fiber membrane permeability, as reported elsewhere [43, 60]. Increases in Hct, Hb, RBC, and WBC after exertion were also observed, corresponding to previous studies [4, 5]. It is well-established that red blood cell parameters rise following splenic contraction to increase oxygen-carrying capacity during intense exercise [4, 5, 61, 62]. Furthermore, increased WBC levels are thought to be due to a shift into the bloodstream of lymphocyte [63] and neutrophil subpopulations [64] from the spleen, bone marrow, and lymph nodes. These reactions are influenced by cortisol and catecholamine releases following physical effort [65, 66].

Although the enhancement of WBC levels immediately after chukkas was similar to changes reported by Zobba et al. [5], a disparity in leucocyte modulation and distribution was detected compared to their study. They demonstrated that a transient rise in WBC was observed immediately after the chukka [5], compared to a more prolonged increased WBC count immediately and until 180 minutes after the chukka in our study. Moreover, the presence of increased lymphocytes, but not neutrophils, in polo ponies of the previous study [5] contrasted with our findings, which showed increased neutrophils instead of lymphocytes. Since a change in leukocyte distribution and magnitude of leucocytosis are related to exercise intensity and duration [67, 68], a difference in effort intensities (indicated by HR during the match) between the previous (~80 beats/min) and current (~175 beats/min) studies may be the underlying reason for the discrepancy. Although the ponies in this study showed increased serum cortisol levels and hematological profile after the chukkas, all parameter changes were within the normal reference range [5, 69]. These laboratory results may reflect the ponies' capability for physiological adjustment during physical effort in polo matches.

Concerning autonomic regulation during exertion, HRV typically decreases during conventional exercise in distinct equestrian sports [13, 39, 40, 70]. A lower HRV indicates a synchronous increase in sympathetic tone and decreased vagal tone or independent action of increased sympathetic or reduced vagal activities [29, 33]. It has been reported that RMSSD decreased during competitive exercise but did not differ between horses in jumping and dressage events [13]. However, a difference in autonomic responses to ultra-short-term stimuli was observed in sports horses. Villas-Boas et al. (2022) demonstrated a lower LF/HF ratio—indicating a minimal autonomic response to the startle test—in endurance horses compared

with dressage and polo horses [53]. It was suggested that autonomic response to startling challenges, but not exercise, differed among horses in different fields. In this study, a decrease in HRV variables derived by time domain (SDNN, RMSSD, and pNN50), frequency domain (VLF, LF, and HF bands), and nonlinear (SD1 and SD2) methods was detected, regardless of field play positions, during exercise in the warm-up and within the chukka. These reductions in HRV were consistent with previous reports describing HRV modulation in response to exercise [22, 38], mirroring a decreased role of the vagal component and a shift toward sympathetic activity during exercise in polo. More importantly, higher HR indicated greater effort intensity during the chukka than during the warm-up period. The increased effort intensity corresponded to a further decrease in HRV variables and, in turn, a progressive decrease in vagal activity during the chukka. These results indicated that exercise intensity was a contributing factor to HRV modification. Even though the frequency domain variables (VLF, LF, HF, and total power bands) decreased during the warm-up and the chukka, the contribution of each power spectrum band to the total power spectrum during the chukka differed from other given periods in the study.

The LF/total power ratio remained unchanged throughout the study. In contrast, the VLF/total power ratio increased, coinciding with a decreased HF/total power ratio during exercise. These results indicated a significant contribution of the VLF band, corresponding to a decreased HF band, to the total power spectrum during exercise. Since the VLF band is modulated in response to vasomotor tone, thermoregulation, and renin-angiotensin action [71], an increased proportion of VLF contribution may, at least in part, point out the apparent effect of vasomotor tone thermoregulation and renin-angiotensin action, along with sympathetic dominance, on autonomic regulation during the chukka. An increase in LF/HF ratio and SD2/SD1 ratios provided supporting evidence of an increased sympathetic and decreased vagal activity during exertion. Although no differences in those HRV variables were observed among field-play positions, it was marked in the stress index. A later reduction in the stress index of the ponies in the defensive position suggested a tendency toward more stress than other positions. This observation is partly consistent with a previous study [8] reporting that defenders had the highest anaerobic power utilization, leading to reduced blood pH compared with those in other positions. Accordingly, more care may be taken in selecting proper ponies for defensive positions to avoid compromised horse welfare in equestrian polo.

Although we report differences in hormonal, hematological, and autonomic responses according to the positions in which ponies compete in the first chukka of a low-goal match, the question arises as to what extent such modulations would occur in ponies competing in more than one chukka. In addition, the effect of age on the effort intensity distribution and HRV modification in such ponies needs further investigation.

This study's main limitation was that different players rode the ponies, and we did not control for their weight. The potential impact of different loadings on physiological responses in polo ponies during the games may be considered, distorting logical results. Since a team's handicap is calculated from a sum of player handicaps irrespective of a similar distribution of individual handicaps within each team, differences in a player's handicap (indicating distinct athletic ability) in a similar position may lead to a significant deviation of the results in each field-play position.

## Conclusion

The effort intensity distribution differed among polo ponies in different field-play positions during the chukka. Blood cortisol levels and hematological parameters did not vary with position. However, cortisol levels increased immediately and 30 minutes after the match. In

contrast, red and white blood cell variables (except lymphocytes) increased immediately to 180 minutes after matches, while CK enzyme levels rose until 150 minutes afterward. A marked decrease in HRV indicated a shift toward sympathetic activity and physiological stress during warm-up and competition until 90 minutes afterward. The increased stress index of ponies at the number 4 position lasted longer than the other positions, implying that ponies in the defensive position tended to experience more stress than other positions.

These results provide insight into the effort intensity distribution, hematology, and autonomic regulation in response to exertion in polo ponies in different field-play positions. This information may, at least in part, be beneficial for selecting appropriate polo ponies regarding adequate fitness and proper autonomic function or adding specific training programs to reach appropriate fitness levels for typical field-play positions in polo, especially the number 2 and 4 ponies. More importantly, compared to before the game, the finding that expression of sympathetic dominance lasts 90 minutes after the match could heighten awareness of the prolonged physiological stress and, in turn, compromised animal welfare in polo ponies during this period, despite playing only one chukka.

## Supporting information

**S1 Fig. The MATLAB MAT-file demonstrates all analysis results of a polo pony participating in the match play, including time-domain result, frequency-domain results nonlinear results and autonomic nervous system indexes.** In addition, this file also includes the HR time series and raw RR interval data.
(DOCX)

## Acknowledgments

We want to thank Harald Link, the president of the Thailand Equestrian Federation (TEF), and Nunthinee Tanner, the vice president, for allowing us to conduct the experiment at the Thai Polo and Equestrian Club. We also thank Surapol Puthapitak and Manu Cereceda, the club's manager, for their tremendous support in facilitating data collection.

## Author Contributions

**Conceptualization:** Kanokpan Sanigavatee, Chanoknun Poochipakorn, Onjira Huangsaksri, Thita Wonghanchao, Metha Chanda.

**Data curation:** Kanokpan Sanigavatee, Chanoknun Poochipakorn, Onjira Huangsaksri, Thita Wonghanchao, Mona Yalong, Kanoknoot Poungpuk, Kemika Thanaudom, Metha Chanda.

**Formal analysis:** Kanokpan Sanigavatee, Chanoknun Poochipakorn, Onjira Huangsaksri, Thita Wonghanchao, Mona Yalong, Kanoknoot Poungpuk, Kemika Thanaudom, Metha Chanda.

**Funding acquisition:** Metha Chanda.

**Investigation:** Kanokpan Sanigavatee, Chanoknun Poochipakorn, Onjira Huangsaksri, Thita Wonghanchao, Mona Yalong, Kanoknoot Poungpuk, Kemika Thanaudom, Metha Chanda.

**Methodology:** Kanokpan Sanigavatee, Chanoknun Poochipakorn, Onjira Huangsaksri, Thita Wonghanchao, Metha Chanda.

**Project administration:** Metha Chanda.

**Resources:** Kanokpan Sanigavatee, Chanoknun Poochipakorn, Onjira Huangsaksri, Thita Wonghanchao, Metha Chanda.

**Software:** Kanokpan Sanigavatee, Chanoknun Poochipakorn, Onjira Huangsaksri, Thita Wonghanchao, Metha Chanda.

**Supervision:** Metha Chanda.

**Validation:** Kanokpan Sanigavatee, Chanoknun Poochipakorn, Onjira Huangsaksri, Thita Wonghanchao, Mona Yalong, Kanoknoot Poungpuk, Kemika Thanaudom, Metha Chanda.

**Visualization:** Kanokpan Sanigavatee, Chanoknun Poochipakorn, Onjira Huangsaksri, Thita Wonghanchao, Mona Yalong, Kanoknoot Poungpuk, Kemika Thanaudom, Metha Chanda.

**Writing – original draft:** Kanokpan Sanigavatee, Chanoknun Poochipakorn, Onjira Huangsaksri, Thita Wonghanchao, Mona Yalong, Kanoknoot Poungpuk, Kemika Thanaudom.

**Writing – review & editing:** Metha Chanda.

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
