## [Decision Letter · Decision Letter 0]

26 Feb 2024

PONE-D-23-41528Hematology, physiology, and autonomic regulation in polo ponies with different field-play positions during low-goal polo matchesPLOS ONE

Dear Dr. Chanda,

Thank you for submitting your manuscript to PLOS ONE. After careful consideration, we feel that it has merit but does not fully meet PLOS ONE’s publication criteria as it currently stands. Therefore, we invite you to submit a revised version of the manuscript that addresses the points raised during the review process.

We look forward to receiving your revised manuscript.

Kind regards,

Aziz ur Rahman Muhammad

Academic Editor

PLOS ONE

Journal Requirements:

**Additional Editor Comments:**

Dear Authors

Reviewers have suggested minor revision especially check grammatical, sentence structure and fluency. Furthermore, also focus on precision in methods section and result.

Reviewers' comments:

Reviewer's Responses to Questions

**Comments to the Author**

1. Is the manuscript technically sound, and do the data support the conclusions?

Reviewer #1: No

Reviewer #2: Yes

Reviewer #3: Yes

2. Has the statistical analysis been performed appropriately and rigorously? 

Reviewer #1: N/A

Reviewer #2: Yes

Reviewer #3: Yes

3. Have the authors made all data underlying the findings in their manuscript fully available?

Reviewer #1: Yes

Reviewer #2: Yes

Reviewer #3: Yes

4. Is the manuscript presented in an intelligible fashion and written in standard English?

Reviewer #1: No

Reviewer #2: Yes

Reviewer #3: Yes

5. Review Comments to the Author

Reviewer #1: I withdraw as I am not expert on horses physiology. I am suggesting rejection as there is no other option to select. You need to send this to people that are dealing this equine physiology. I have done some editing but I quit.

Reviewer #2: I would thanks the authors for the effort and for the best quality of the manuscript. However, there is some mistakes to correct them.

General comments:

- The authors need to work with a native English speaker/writer to correct grammatical and punctuation errors throughout the manuscript.

- Title of paper need to be reformulated

Abstract

- The aim of the study is not clear

- Conclusion need to be reorganized

Introduction

- More details and information about Polo ponies

- L64-L68: Reformulate this paragraph

- L126-L129: Reformulate this paragraph

- The problematic is not clear

Methods

- Inclusion and exclusion criteria are not clear

- More details and information are needed in the experimental protocol

Results

- Figure 2 is not very clear.

Discussion

- L448-L449: Add reference

- L451-L454: Add reference

- L457-L463: References are needed

- L464-L470: References are needed

- L528-L531: References are needed

- L540-L544: There’s no consistence between paragraphs

-

- Discuss limitations of the study, taking into account sources of potential bias or imprecision. Discuss both direction and magnitude of any potential bias

- Discuss the generalisability (external validity) of the study results

- Discuss the practical implications and future research

Reviewer #3: The authors present a well written manuscript on the physiological and stress effects of chukkas in polo ponies. Furthermore, authors analyse the difference in these horses depending on their playing fields. This study is of importance in the current era of animal welfare, and understanding of equine sports. Therefore, I recommend the publication of this study in PLOS One, as it fits to the scope of the journal. I suggest minor revisions, please see hereunder:

Line 95-114: please shorten this text, as not all variables need to be introduced here.

Line 123: if I understal well, reference 38 is not accepted for publication yet. I suggest authors to use https://doi.org/10.1016/j.jevs.2021.103716 which showed a decrease of time domain HRV parameters with exercise. Authors can also add https://doi.org/10.3389/fvets.2022.939534 which showed a decrease in HRV during groundwork.

Line 143-4: please review decimals: 1 decimal for age and 0 for weight seem more appropriate (most weighting instruments are not accurate at the kg in horses, and hundreds of years are not easy to relate in age)

Line 177: could authors convert ukat/l to international units

Line 180-181: please add https://doi.org/10.1016/j.jevs.2021.103716 as reference for the use of this instrument during exercising horses

Line 231-234: please indicate if the conditions for applying ANOVA were met for the two-grouping variables.

Along the document: I suggest replacing ‘number four players’ by ‘defenders’ and ‘number one players’ by ‘forwarders’, and number two and three by ‘mildfielders’

Line 264-266: could authors specify why number 2 and number 3 players were not merged for analysis, as they do play as the same role in this discipline. If not, why is there a difference in results in number 2 and number 3 players? Was this expected? If yes, why? If not, what interpretation can authors give to the difference?

Line 284: The increase in CK is significant statistically, but it does not seem to be clinically relevant. Please convert to international units.

Line 308: HR at 150 min is not significantly lower than baseline, according to authors’ criteria (p<0.05).

Line 320-324 and Fig 3: RR intervals and HR are quite the same data, expressed differently. I suggest removing one of them (RR intervals in my opinion) to share a more to the point information.

Line 482: see above, I do not believe the increase, despite it statistical significance, is clinically relevant.

Major

Line 204-213: I do not believe all HRV variables need to be used in this protocol. Different variables show the same component of HRV, in different ways. Some of these have not been validated in (exercising) horses. Showing all variables results in repetitive results about different variables. I suggest authors focus on short-term HRV. Adapt the Introduction and Discussion sections accordingly.

6. PLOS authors have the option to publish the peer review history of their article (what does this mean?). If published, this will include your full peer review and any attached files.

Reviewer #1: No

Reviewer #2: **Yes: **Jamel Halouani

Reviewer #3: No

---

## [Author Response · Author response to Decision Letter 0]

26 Mar 2024

PONE-D-23-41528

Hematology, physiology, and autonomic regulation in polo ponies with different field-play positions during low-goal polo matches

Dear reviewers

We’d like to thank the reviewers for their valuable time reviewing our work. We have gone through the reviewer's comments before addressing all points. The corrections and modifications are highlighted in green.

Additional Editor Comments:

Dear Authors

Reviewers have suggested minor revision especially check grammatical, sentence structure and fluency. Furthermore, also focus on precision in methods section and result.

Response to the editor

We’d like to thank the editor and reviewer for this comment. In fact, the MS has been grammatically corrected by the Cambridge proofreading service (as an attached certificate). However, the revised MS will be grammatically corrected by Cambridge proofreading again before resubmission.

Reviewer #2: I would thanks the authors for the effort and for the best quality of the manuscript. However, there is some mistakes to correct them.

General comments:

- The authors need to work with a native English speaker/writer to correct grammatical and punctuation errors throughout the manuscript.

Response to the reviewer

We’d like to thank the reviewer for this comment. In fact, the MS has been grammatically corrected by the Cambridge proofreading service (as an attached certificate). However, the revised MS will be grammatically corrected by Cambridge proofreading again before resubmission.

- Title of paper need to be reformulated

Response to the reviewer

We’d like to thank the reviewer for this comment. The title was modified to be short and concise: “Hematological and physiological responses in polo ponies with different field-play positions during low-goal polo matches.”

Abstract

- The aim of the study is not clear

- Conclusion need to be reorganized

Response to the reviewer

We’d like to thank the reviewer for this comment. A brief background and rationale for the study were added to the abstract on page 2, lines 24-25. The conclusion was modified slightly on page 2, lines 43-45.

Introduction

- More details and information about Polo ponies

Response to the reviewer

We’d like to thank the reviewer for this comment. We’ve modified the text to add more information about polo ponies and their training patterns on page 3, lines 61-70.

- L64-L68: Reformulate this paragraph

Response to the reviewer

We’d like to thank the reviewer for this comment. We’ve reformulated this paragraph on pages 3-4, lines 71-76.

- L126-L129: Reformulate this paragraph

Response to the reviewer

We’d like to thank the reviewer for this comment. We’ve reformulated this paragraph on page 5-6, lines 115-123.

- The problematic is not clear

Response to the reviewer

We’d like to thank the reviewer for this comment. We’ve revised this paragraph on page 6, lines 124-125.

Methods

- Inclusion and exclusion criteria are not clear

Response to the reviewer

We’d like to thank the reviewer’s comment. We’ve revised and added more information regarding the inclusion and exclusion criteria on page 6, lines 139-143.

- More details and information are needed in the experimental protocol

Response to the reviewer

We apologize for giving the reviewer unclear information. Typically, the experimental protocol is on pages 7-8, lines 147-218, and includes the exercise regimen for polo, hormonal and hematological analysis, HRV analysis, effort intensity zone determination, and speed/cover distance during chukka. We’ve reorganized the subheadings in the experimental protocol to provide more precise information.

Results

- Figure 2 is not very clear.

Response to the reviewer

We do apologize for not knowing which part of Figure 2 remains unclear for the reviewer. Anyway, we will explain the information in Figure 2 a little bit. In this figure, we measured HR during the chukka period and categorized it into 5 zones, based on the %HRmax as stated on page 9, lines 207-214. The two-way ANOVA to evaluate the interaction effect of field play position and HR zone in which the quantitative measurement is the time spent within each zone. We found the interaction effect of field play positions and HR zones on time spent within the zones. The different letters in each field play position meant statistical significance between pairs of comparisons, as shown in Figure 2. We also found the difference between time spent in zone 2 of position 2 (dark blue bar graph) and 4 (black bar graph) and time spent in zone 4 of positions 2 and 4. Based on these results, we suggest that polo ponies at different field-play positions experienced different levels of effort intensity.

Discussion

- L448-L449: Add reference

Response to the reviewer

We’d like to thank the reviewer’s comment. We’ve added the reference according to the comment on page 17, line 407. 

- L451-L454: Add reference

Response to the reviewer

We’d like to thank the reviewer’s comment. We’ve added the reference according to the comment on page 18, line 411.

- L457-L463: References are needed

Response to the reviewer

We’d like to thank the reviewer’s comment. We’ve added the reference according to the comment on page 18, line 424.

- L464-L470: References are needed

- L528-L531: References are needed

Response to the reviewer

We’d like to thank the reviewer’s comment. In fact, the documents regarding the 2 above comments refer to the result of this study (page 18, lines 419-423) that may be consistent with the previous report (contributors W. Polo. In: Wikipedia, editor. The Free Encyclopedia; https://en.wikipedia.org/wiki/Polo). In this case, we’ve added this reference on page 18, line 419 and 424, accordingly.

- L540-L544: There’s no consistence between paragraphs

Response to the reviewer

We’d like to thank the reviewer’s comment. We’ve modified it to be more precise on page 21, lines 495-500.

- Discuss limitations of the study, taking into account sources of potential bias or imprecision. Discuss both direction and magnitude of any potential bias

Response to the reviewer

We’d like to thank the reviewer’s comment. We’ve modified it and discussed more in the study’s limitation on page 21-22, lines 506-512.

- Discuss the generalisability (external validity) of the study results

We’d like to thank the reviewer’s comment. In fact, we are uncertain that I understand the comment well. If my understanding is correct, I would explain that the use of heart rate and heart rate variability have been proven reliable parameters to indicate exercise intensity and stress response, as mentioned in the introduction and discussion. Moreover, the use of a heart rate monitoring (HRM) device for HRV determination was added on page 8, lines 175-176. Since there is no report of HRV measurement in polo ponies during the actual competition (chukka), we adopted this technique and device due to its easy-to-use characteristic and no interference with the horse’s movement during fierce competition. However, based on another reviewer’ comment to select only the parameter to indicate short-term variation of the heart rate. Eventually, The TINN, RR triangular index, DC, DC mode and ANS index (SD1%, SD2%, PNS and SNS indices) were removed from the results and other parts of the manuscript.

- Discuss the practical implications and future research

We’d like to thank the reviewer’s comment. Regarding the practical implication, we have mentioned it already in the conclusion part on pages 22, lines 523-531.

 

Reviewer #3: The authors present a well written manuscript on the physiological and stress effects of chukkas in polo ponies. Furthermore, authors analyse the difference in these horses depending on their playing fields. This study is of importance in the current era of animal welfare, and understanding of equine sports. Therefore, I recommend the publication of this study in PLOS One, as it fits to the scope of the journal. I suggest minor revisions, please see hereunder:

Line 95-114: please shorten this text, as not all variables need to be introduced here.

We’d like to thank the reviewer’s comment. We’ve modified and removed the general information of HRV parameter from the introduction part accordingly.

Line 123: if I understal well, reference 38 is not accepted for publication yet. I suggest authors to use https://doi.org/10.1016/j.jevs.2021.103716 which showed a decrease of time domain HRV parameters with exercise. Authors can also add https://doi.org/10.3389/fvets.2022.939534 which showed a decrease in HRV during groundwork.

We’d like to thank the reviewer’s comment. Since we’ve modified the introduction part as above comment, this context mentioned in line 123 has been removed from the introduction part.

Line 143-4: please review decimals: 1 decimal for age and 0 for weight seem more appropriate (most weighting instruments are not accurate at the kg in horses, and hundreds of years are not easy to relate in age)

We’d like to thank the reviewer’s comment. We’ve modified it on page 6, lines 131-132.

Line 177: could authors convert ukat/l to international units

We’d like to thank the reviewer’s comment. Since the Plos One journal guideline indicates the use of an SI unit rather than an international unit. So, we continued using ukat/l as the SI unit instead of the international unit (IU/L) for the determination of CK level in this manuscript.

Line 180-181: please add https://doi.org/10.1016/j.jevs.2021.103716 as reference for the use of this instrument during exercising horses

We’d like to thank the reviewer for his comment. We’ve modified the context and added these references on page 8, lines 175-176.

Line 231-234: please indicate if the conditions for applying ANOVA were met for the two-grouping variables.

We’d like to thank the reviewer for his comment. We apologize for giving the reviewer the wrong message. In fact, it should have been: “In an analysis of two independent variables” (field-play position and time), we measured the interaction effect of field-play position-by-time and the independent effect of each variable. We’ve modified it on page 10, lines 221-223.

Along the document: I suggest replacing ‘number four players’ by ‘defenders’ and ‘number one players’ by ‘forwarders’, and number two and three by ‘mildfielders’

We’d like to thank the reviewer for his comment. In fact. the different field-play positions play a distinct role during the game, particularly in the midfielder played by the number 2 and 3 position, as stated on page 18, lines 413-419. To state that only midfielders may not convey the message of different stress responses in player no. 2 and 3. However, we’ve referred to the player numbers (1-4) and roles during the game (forwarder, midfielder and defender) already on page 7, lines 150-152. We hope that this context can convey the meaning of player position and role during the game.

Line 264-266: could authors specify why number 2 and number 3 players were not merged for analysis, as they do play as the same role in this discipline. If not, why is there a difference in results in number 2 and number 3 players? Was this expected? If yes, why? If not, what interpretation can authors give to the difference?

We’d like to thank the reviewer for his comment. Although numbers 2 and 3 are regarded as midfielders in the polo game, their roles are a bit different, as stated on page 18, lines 413-419. The team manager's trainer will assign the players who fit those field-play positions. Since there are no reports of stress responses on horses in these positions, our result proved that horses with different positions perform different exercise intensities, particularly in no. 2 and 4. So horses no. 2 and 3 cannot be merged for analysis.

Line 284: The increase in CK is significant statistically, but it does not seem to be clinically relevant. Please convert to international units.

We’d like to thank the reviewer’s comment. Since the Plos One journal guideline indicates the use of an SI unit rather than an international unit. So, we continued using ukat/l as the SI unit instead of the international unit (IU/L) for the determination of CK level in this manuscript.

Line 308: HR at 150 min is not significantly lower than baseline, according to authors’ criteria (p<0.05).

We’d like to thank the reviewer’s comment. We’ve removed the words “150 (p = 0.0728)” from the main text accordingly.

Line 320-324 and Fig 3: RR intervals and HR are quite the same data, expressed differently. I suggest removing one of them (RR intervals in my opinion) to share a more to the point information.

We’d like to thank the reviewer for his comment. In fact, HR shows a strongly negative correlation to RR intervals, meaning that increased HR corresponds to decreased RR intervals and vice versa. As they reflect the influences of both sympathetic and vagal components on the cardiac cycle; so, neither HR nor RR interval can be removed from the main text.

Line 482: see above, I do not believe the increase, despite it statistical significance, is clinically relevant.

We’d like to thank the reviewer for his comment. We totally agree with the reviewer that an increased CK is not clinically relevant as “The observed increase in CK enzyme levels parallels those of a previous report [5], demonstrating a rise in after the chukkas, but within the normal reference range of the enzyme” on page 19, lines 437-438. The latter sentence as “Increased CK levels are believed to accompany high-intensity exercise due to muscle microtrauma, leading to increased fiber membrane permeability, as reported elsewhere [43, 60]” was informed to provide the reason how it increased (but within normal range) following the polo game.

Major

Line 204-213: I do not believe all HRV variables need to be used in this protocol. Different variables show the same component of HRV, in different ways. Some of these have not been validated in (exercising) horses. Showing all variables results in repetitive results about different variables. I suggest authors focus on short-term HRV. Adapt the Introduction and Discussion sections accordingly.

We’d like to thank the reviewer’s comment. We’ve already removed 1) the time-domain HRV parameters, including TINN, RR triangular index, DC, DCmod, and 2) the ANS index, including SD1%, SD2%, PNS, and SNS indices from the main text, as the remaining parameter can demonstrate a clear result accordingly.

---

## [Decision Letter · Decision Letter 1]

18 Apr 2024

Hematological and physiological responses in polo ponies with different field-play positions during low-goal polo matches

PONE-D-23-41528R1

Dear Dr. Chanda,

We’re pleased to inform you that your manuscript has been judged scientifically suitable for publication and will be formally accepted for publication once it meets all outstanding technical requirements.

Kind regards,

Aziz ur Rahman Muhammad

Academic Editor

PLOS ONE

Additional Editor Comments (optional):

Dear Authors

Thanks for revision. Good Luck

Reviewers' comments:

Reviewer's Responses to Questions

**Comments to the Author**

1. If the authors have adequately addressed your comments raised in a previous round of review and you feel that this manuscript is now acceptable for publication, you may indicate that here to bypass the “Comments to the Author” section, enter your conflict of interest statement in the “Confidential to Editor” section, and submit your "Accept" recommendation.

Reviewer #2: All comments have been addressed

Reviewer #3: All comments have been addressed

2. Is the manuscript technically sound, and do the data support the conclusions?

Reviewer #2: Yes

Reviewer #3: Yes

3. Has the statistical analysis been performed appropriately and rigorously? 

Reviewer #2: Yes

Reviewer #3: Yes

4. Have the authors made all data underlying the findings in their manuscript fully available?

Reviewer #2: Yes

Reviewer #3: Yes

5. Is the manuscript presented in an intelligible fashion and written in standard English?

Reviewer #2: Yes

Reviewer #3: Yes

6. Review Comments to the Author

Reviewer #2: I would thank the authors for the best quality of this manuscript in this corrected version. I think that the manuscript is well corrected and now is suitable for publication

Reviewer #3: The authors have answered all questions and adapted accordingly the manuscript. Thanks to the authors for this revised manuscript, which is in my opinion suitable for publication.

7. PLOS authors have the option to publish the peer review history of their article (what does this mean?). If published, this will include your full peer review and any attached files.

Reviewer #2: **Yes: **Jamel Halouani

Reviewer #3: No

---

## [Editor Report · Acceptance letter]

29 Apr 2024

PONE-D-23-41528R1 

PLOS ONE

Dear Dr. Chanda, 

I'm pleased to inform you that your manuscript has been deemed suitable for publication in PLOS ONE. Congratulations! Your manuscript is now being handed over to our production team.

Kind regards, 

on behalf of

Dr. Aziz ur Rahman Muhammad 

Academic Editor

PLOS ONE